# Vessel Trajectory Analysis in Designated Harbor Route Considering the Influence of External Forces

**Ho Namgung** [1] and **Joo-Sung Kim** [2,*]

1 Division of Naval Officer Science, Mokpo National Maritime University, Mokpo 58628, Korea; ngh2009@mmu.ac.kr

2 Division of Navigation Science, Mokpo National Maritime University, Mokpo 58628, Korea

* Correspondence: jskim@mmu.ac.kr

**Abstract:** A vessel must navigate along designated routes within a harbor area to ensure navigation safety. The impact of strong currents is one of the most dangerous factors in coastal navigation. However, it is challenging to determine the deviation of a ship in advance from the ship's position data in the case of a marine accident. In this study, to support the decision-making of ship navigators and vessel traffic service (VTS) operators in track monitoring tasks, tracks were classified according to the tidal stream, and the track distribution was analyzed according to the tidal current situations. Marine accident analysis was performed to investigate the tidal influence on ship tracks. Track data were collected for 12 months from a VTS center in Korea, and tidal information was collected through a meteorological observation buoy. Representative tracks were extracted from the track data using the support vector regression (SVR) seaway model. K-fold cross-validation and a grid search were performed to determine the optimal parameters. The ship tracks appeared in specific patterns according to the forces and directions of tidal currents, and specific deviation patterns were observed. This study is expected to contribute to the reduction of marine accidents by predicting ship trajectories according to the tidal situations in advance.

**Keywords:** vessel traffic services; vessel trajectory analysis; anomalous behavior; support vector machine; machine learning; SVR seaway model; marine accident

## 1. Introduction

In recent years, it has become possible to use ship navigational data and real-time marine environmental data for traffic analysis, with the advancement of maritime information and communication technology. In addition, the results of marine traffic analysis contribute to the development of the vessel traffic services (VTS) monitoring system and navigational decision-making equipment applications, which are used for the prevention of potential marine accidents. Despite advances in technology and efforts, marine accidents have continued to occur. According to a report by the Korean Maritime Safety Tribunal (KMST), a total of 20,064 marine accidents occurred from 2010 to 2019, of which 2371 occurred in the waters near ports, including harbors, and in the waterways entering ports [1]. With the enlargement of vessels, a large amount of cargo can be transported at lower costs. However, in the event of a marine accident, the scale of the marine accident is also larger, causing serious environmental pollution and enormous economic loss along with the loss of human life. Therefore, to prevent potential marine accidents and maximize the efficiency of ship navigation, it is recommended that ships be navigated through designated routes in the port and its approaching waters, and a VTS system be installed and operated. VTS has been established by authorized governments in necessary areas to promote safe and efficient navigation and prevent marine accidents according to the International Convention for the Safety of Life at Sea 1974 (SOLAS 1974)

Rule 12 in Chapter 5 and the International Maritime Organization (IMO) RESOLUTION A.857 (20) on Guidelines for Vessel Traffic Services [2,3].

For cases wherein a ship navigates an area with a high risk of accidents or with sensitive conditions for navigation, such as a route within a port limit and its approaching areas, the master or navigator checks the location of the ship and confirms whether the vessel maintains an appropriate route. Maintaining the proper course and speed to be taken by the ship at a specific location is the most basic means of navigation for the ship to maintain its designated route. The relative position of the ship in the near future with respect to the current location is determined on the basis of the engine operation and course selection. However, the ship position in the near future is affected not only by the course selection and speed control, but also by the navigational environment in which the ship is located. The depth of the route and the width of the waterway can be predicted in advance through information inferred from the nautical chart; however, the effects of traffic conditions and marine environment, such as tidal currents and winds, are difficult to predict solely on the basis of the observation and confirmation of the ship's navigator.

A ship's dead reckoning position (DRP) has traditionally been used as a method for predicting the position of a ship [4]. DRP is used to predict the navigational relationship with other vessels and the situation when the ship encounters situations in traffic that are considered to be dangerous. In addition, DRP is used to know the position of the ship in question or another ship after a certain time. In an open sea with a large margin of water, the position of the ship in the near future can be determined from the current course and speed of the ship, assuming that there is no change in the track of the ship. However, this type of method cannot be used in waters within harbors where there are frequent variations in course and speed to keep track of the designated route. To address these problems, a modified DRP calculation method reflecting the ship's planned route was proposed [5,6]. In addition, as machine learning is used for route prediction, a method for predicting the ship's position by learning the ship's trajectory data was proposed [7]. However, the aforementioned methods have a limitation in utilizing location-based automatic identification system (AIS) data, which do not consider the navigational environment of the ship. In the conventional prediction method, DRP is predicted using the ship's current position, speed, and course [4], and the DRP calculation reflecting the track does not consider the ship's external forces [6]. Therefore, it is necessary to analyze the influence of external forces with extracted sailing routes in calculating the ship's DRP or estimating the ship's future positions. Even if a ship navigates an identical route, factors that affect the ship's movement, such as the directions and tidal stream forces, differ depending on the situation and the time at which the ship sails [8]. In particular, the ship can be made to drift by the influence of strong currents in areas with large tidal differences. The ship's drift is one of the causes of marine accidents, such as swerving from the designated route and striking aground.

In this study, we aimed to analyze the effect of external forces on the trajectory formation of a vessel and propose a method that can be used by navigators and VTS operators according to the case of an accident of a vessel that has run aground in the limits of a harbor. The target ship was selected from among the cases of marine accidents for trajectory analysis, and the data were collected for 12 months. In addition, tidal information was collected through a meteorological observation buoy, and the track dataset was classified according to tidal directions and forces. The classified track datasets were reclassified into current situations, such as flood current, ebb current, and near slack, according to the directions and forces of the tide. Machine learning was performed on the track data, and representative tracks were extracted. In addition, a method was proposed to support the decision-making of navigators and VTS operators by selecting a reference point in the designated route of the target area, analyzing the degree of deviation of the track according to each tidal situation, and deriving a linear equation for the distribution of the ship's deviations.

## 2. SVR Seaway Model

The support vector regression (SVR) seaway model is based on SVR, which is a representative machine learning methodology. It is based on the assumption that a ship's track is a passage with a certain pattern when the ship navigates along the designated route, and the start and end points of the ship's leg are identical [9]. SVR utilizes the method of structural risk minimization, which minimizes the probability of error in data with a fixed but unknown probability distribution, while traditional pattern recognition techniques are based on an empirical risk minimization method using a statistical pattern recognition method according to the data distribution [10,11].

The SVR seaway model is a method of learning datasets collected and processed using SVR, consequently extracting a representative trajectory model [9]. Because the method defines a representative route on the basis of a ship's classified track dataset by selecting a certain period, it can be applied to complex marine traffic environments, such as within a port and adjacent waters. The characteristic of machine learning through SVR is to determine the output model by selecting the number of support vectors (SVs) to be learned according to the selected parameter [12,13]. The route extraction method of the SVR seaway model was used in this study because it is suitable for application to complex port waters with frequent changes in navigation patterns due to factors such as port development and changes in navigational environment. Since this method can analyze patterns using navigational datasets for a specific period that requires learning through data and obtain the output from the learned result, it is possible to extract a track model reflecting the navigational characteristics during specific periods. The k-fold cross-validation and grid search techniques were used to select the most important kernel function and optimal parameters in constructing the SVR model for extracting the representative tracks. This technique is derived from the Library for Support Vector Machines (LIBSVM) algorithm proposed by Hsu et al. [14]. This method has been widely used since its proposal and has been recognized for its high reliability. It is also used in constructing the optimal parameter combinations in various machine learning algorithms [15–17]. The process of extracting the representative track model through the SVR seaway model is illustrated in Figure 1.

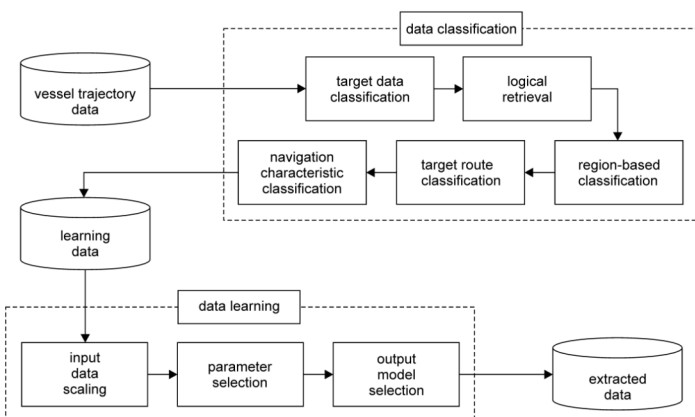

**Figure 1.** Support vector regression (SVR) seaway model algorithm.

First, the ship's navigational data are collected and classified on the basis of the target ship and area. Data for individual ships are converted into a data structure and classified according to the sailing section (leg) to form datasets to serve as input feed for the machine learning algorithm. The track data used for learning are reclassified, and the curved and straight sections are defined by discriminating the curvature of the track distribution and extracting the veering points. The reclassified data are divided into sub-datasets, which form the final dataset for learning and training. For the process of learning, the final track data are selected through data scaling and parameter selection. From the data learning, the final representative track model is extracted, and the database of the track model is constructed. This method of pattern extraction can be used for predicting a ship's track and

determining any anomaly in the behavior. However, most route pattern recognition methods extract the track model according to the ship's path and target leg. In these methods, there is a limitation that the influences of external forces are not considered. In this study, using the SVR seaway model proposed previously, the ship's track pattern is classified and learned according to the type of tidal stream, and a representative track model is extracted for each tidal current situation.

## 3. Methods

The track data were collected by classifying the fusion track data of the AIS and radar images collected at the Incheon Port VTS center, Republic of Korea. The tidal data provided by the Korea Hydrographic and Oceanographic Agency (KHOA) were also used [18]. The KHOA tide information comprises data predicted in real time and the actual measurement data collected every 30 min. In this study, the data provided from a meteorological observation buoy every 30 min were converted into units of seconds through curve fitting using a spline function. The tidal information was used at the time when the target ship passed through the designated area (gate line).

In general, the tidal movement occurs as shown in Figure 2a; the intensity of the tide changes simultaneously, as shown in Figure 2b. Meanwhile, the direction of the tide follows the pattern shown in Figure 2c.

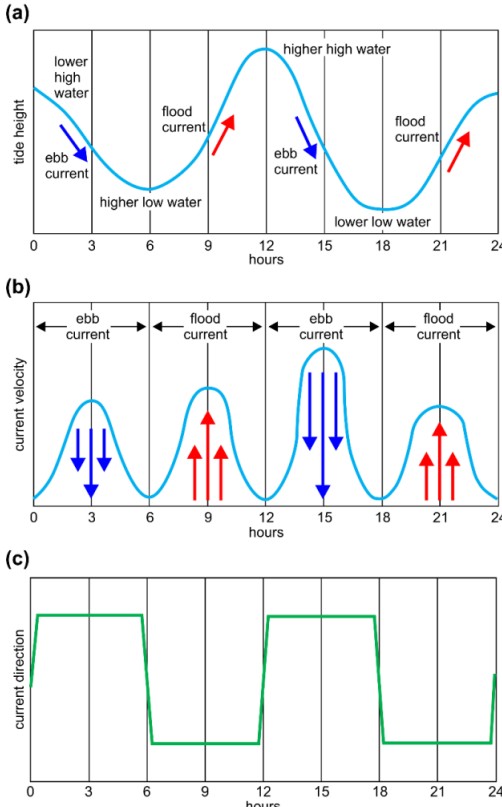

**Figure 2.** General relationship and alteration between tide and tidal streams: (**a**) general tidal movement; (**b**) change in current velocity; (**c**) change in current direction.

As shown in Figure 3, the track dataset was classified into three cases (flood current, ebb current, and near slack), according to the direction of the current and intensity of the tide when the target ship passes through the preset gate line.

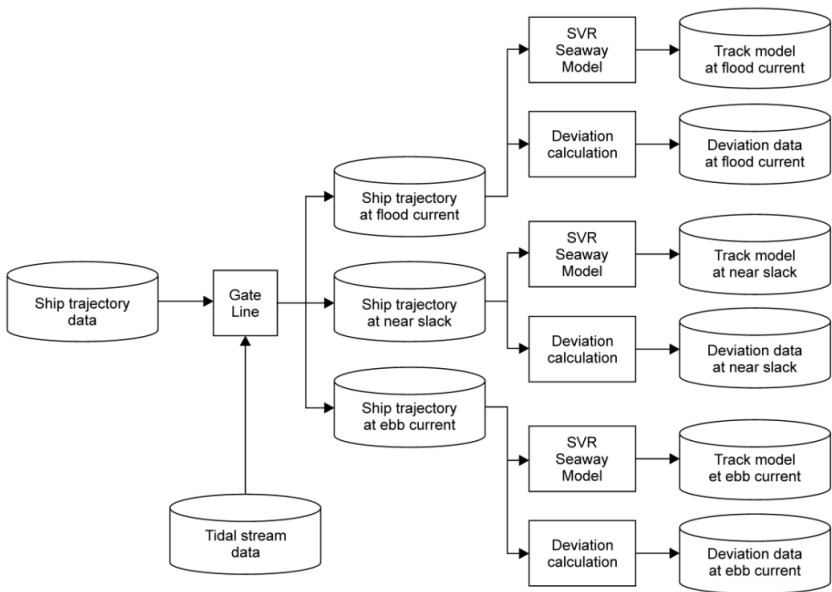

**Figure 3.** Data learning and classifying process reflecting the results of tidal currents.

As shown in Figure 3, the collected track data are classified on the basis of the tidal data and each sub-dataset is extracted from the track model in accordance with the track extraction method of the SVR Seaway Model. The classified tracks are analyzed by calculating the deviated distance from the reference point of the nautical mark in the curved area and by analyzing the relationship of the deviation range of the target ship with the tide intensity for each tide.

## 4. Simulation and Results

The simulation environment was the designated route of Incheon Port, Korea. Incheon Port is the entrance of Seoul, which is the capital city, and the harbor is the second largest international trade port in Korea. Incheon Port consists of Inner Port, Northern Port, Southern Port, Coastal Port, and New Port. There are many large and small islands scattered around this port, which form natural walls. Thus, there is a reduction in wind and waves in this area. However, the tidal range is severe and reaches up to 10 m [19]. The flood current flows to the north-northwest, and the ebb current flows to the south and south-southwest. The speeds of flood current and ebb current are similar. The flood current turns its direction around 0.1–0.8 h after the low water and lasts for about 6.5–6.6 h until 0.5–1.1 h after the high water. The maximum flood current is about 2.0–2.7 knots in the annual mean spring tide rise and occurs around 1.6–2.3 h before the high water. The ebb current turns its direction around 0.5–1.1 h after the high water and last for about 5.8–5.9 h until 0.1–0.8 h after the low water. The maximum ebb current occurs around 2.3–3.3 h before the low water, and the maximum current speed in the annual mean spring tide rise reaches up to 2.0–2.7 knots [19]. The target area in Figure 4 (the black square) shows the area where 27 cases of route swerving were reported in 2018 [20].

The ship passes between the bridge posts and navigates through the curved area to the berth at the scheduled pier in the target area. In particular, according to KHOA's West Coast of Korea Pilot, the target area is a region where the tide moves strongly with a maximum current of 2.7 knots [19], and a maximum of 2.1 knots was recorded in the data collected by the meteorological observation buoy. The route in the area is presented in nautical charts and publications as a sea area where there are obstacles to navigation such as shallow waters and small islands on both sides of the curve [20]. Vessels navigating the area frequently change their course and speed in curves. Therefore, the target area provides an appropriate navigational environment for analyzing the proposed tidal effects. Meanwhile, the tide information was processed and used as shown in Figure 5. The meteorological observation buoy is located 0.78 nautical miles from the reference point. To classify the track dataset,

the gate line was set at the Young-jong Great Bridge in Figure 4, and the track classification criteria were based on the tide information at the time the target ship passed through the gate.

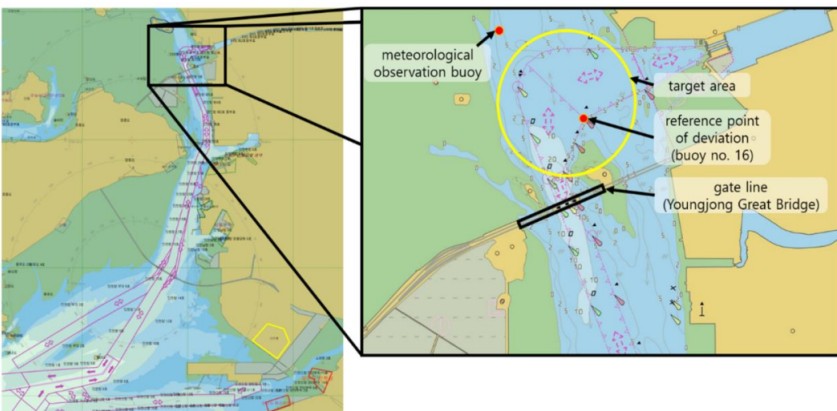

**Figure 4.** Target area and reference points.

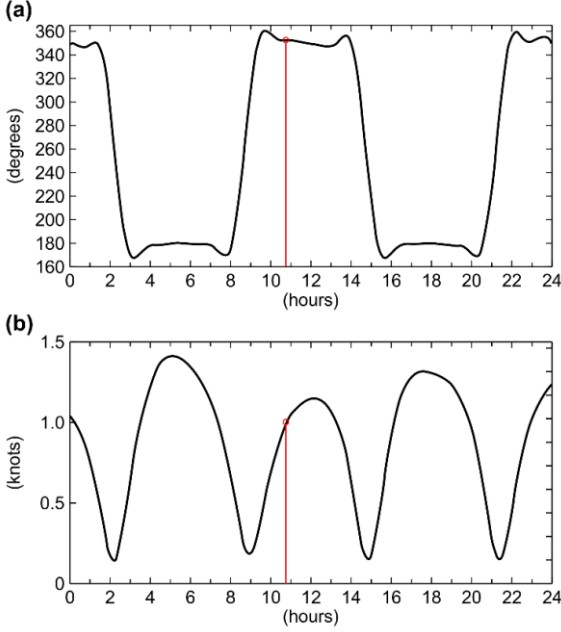

**Figure 5.** Results of tide information analysis: (**a**) changes in current direction; (**b**) changes in current velocity.

Meanwhile, the direction and intensity of the tide when the target ship passes through the gate line can be derived from the data analysis shown in Figure 5. For instance, if the ship passes through the gate line on 14 November 2018 at 10 h 45 min 49 s, from the analysis result of the tide observation data, we can infer that the direction of the tide is 352.4° and the force is 1.0 knot. In this case, the track data are classified as the flood current dataset. The entire track data were classified according to the proposed method and divided into three types of tides, as shown in Figure 6.

Figure 7 compares the track data for each tidal current situation. The representative tracks were extracted from the learned data of the tracks of each classified dataset, as shown in Figure 8. The track data classified according to the direction of the tide showed a specific pattern, as presented in Figure 8.

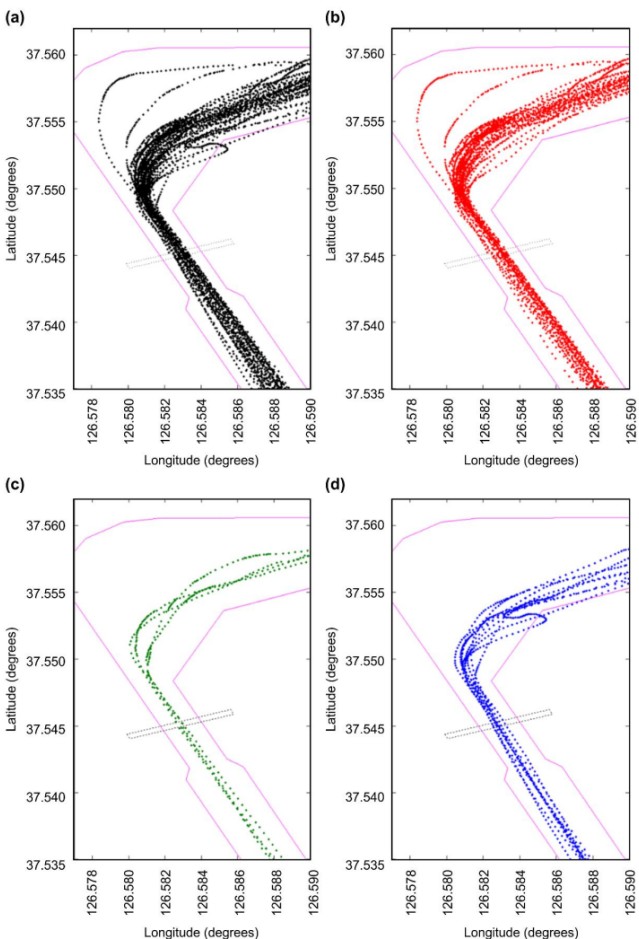

**Figure 6.** Track distribution according to tidal current: (**a**) all current situations; (**b**) flood current; (**c**) near slack; (**d**) ebb current.

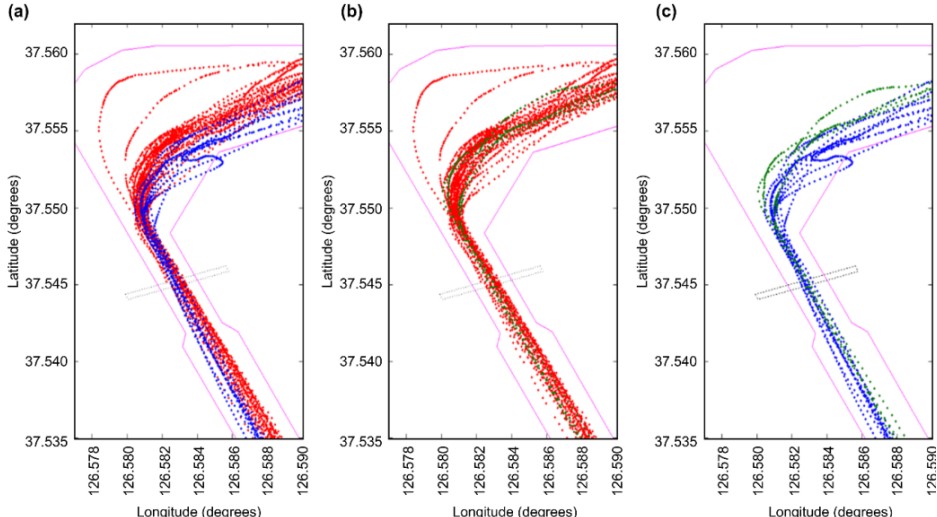

**Figure 7.** Track comparisons according to the tide situation: (**a**) flood current vs. ebb current; (**b**) flood current vs. near slack; (**c**) near slack vs. ebb current.

The distribution of the track and tide intensity at the time of the accident is shown in Figure 9. As shown in Figure 9, two cases of veering off the designated route and one case of a stranding accident were found, and the tide force was 2.1 knots, which was the highest recorded value for that year.

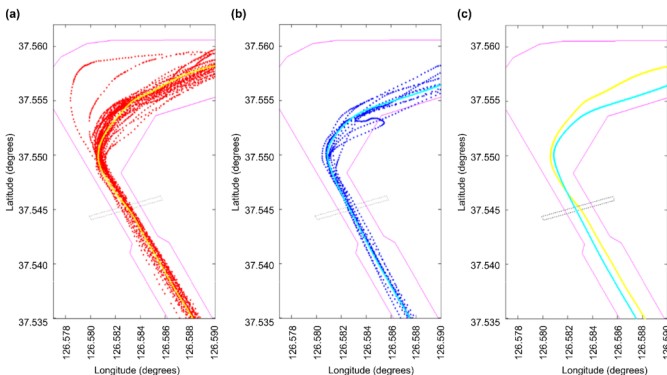

**Figure 8.** Representative track model extraction results: (**a**) flood current; (**b**) ebb current; (**c**) flood current vs. ebb current.

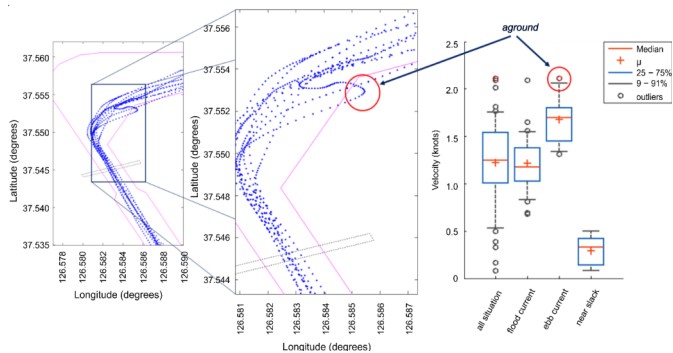

**Figure 9.** Observed tracks and tide intensity of the accident situation.

Figure 10 shows the track deviation from the reference point for each tide situation. The reference point is buoy no. 16 with the reference coordinates of latitude 37° 33.22′ north (N) and longitude 126° 35.11′ east (E). Deviation is defined as the distance from the reference point according to the intensity of the tide. In the flood current, the maximum deviation was 618.57 m, the minimum was 44.04 m, and the average was 218.15 m. In the ebb current, the maximum deviation was 151.30 m, the minimum was −83.43 m, and the average was 96.20 m.

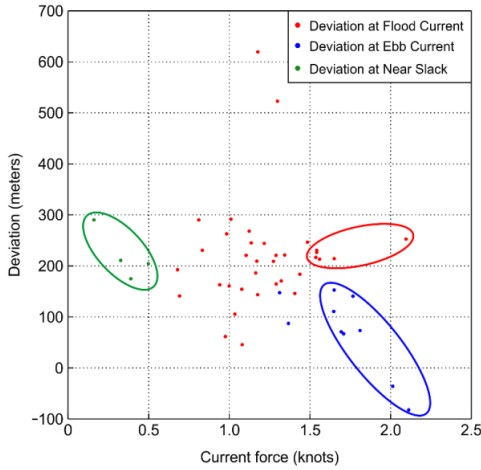

**Figure 10.** Distribution of tidal strength and deviation at all situations.

Figure 11, Table 1, and Table 2 summarize the analysis results of the deviation of the target ship from the reference point when the tide intensity is over 1.5 knots. A simple linear regression analysis was performed to investigate the effect of the force of the current on ship deviation in flood

current. The results show that the force of the current has a significant effect on the ship deviation ($p = 0.040 < 0.05$), and, as the tidal current increases ($B = 58.057$), the ship tracks are formed outside the route. The coefficient of determination ($R^2$), where the force of the current determines the ship's deviation, is 69.3%. Meanwhile, in the case of ebb current, the intensity of the tide has a significant effect on the ship's deviation ($p = 0.002 < 0.01$), and, as the intensity of the current increases ($B = -434.499$), the ship tracks are formed inside the route. The coefficient of determination ($R^2$) is 83.1%. In the case of flood current, as the intensity of the tide increased, the pattern formed on the outer side of the route was detected, while, in the case of ebb current, the pattern formed on the inner side of the route was detected. The simulation results of the proposed method were obtained from a specific ship in a specific sea area on the west coast of Korea. Therefore, in order to apply the proposed method to other cases, it should be verified through various navigational environments and ships in multiple sea areas.

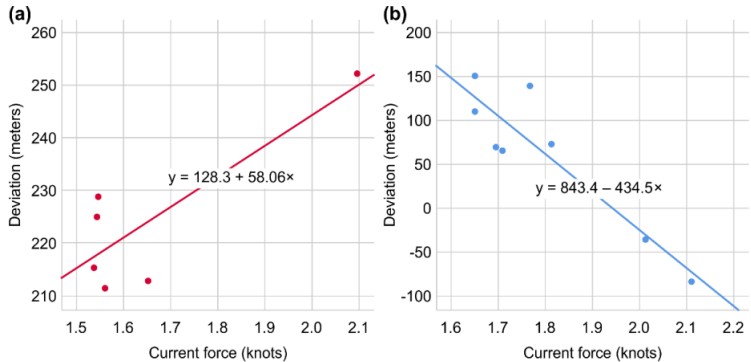

**Figure 11.** Relationship between track deviation and current force: (**a**) flood current; (**b**) ebb current.

**Table 1.** Relationship between track deviation and current force in flood current. SE, standard error.

|  | B | SE | β | t | p |
|---|---|---|---|---|---|
| (Constant) | 128.329 | 32.242 |  | 3.980 | 0.016 |
| Current Force | 58.057 | 19.340 | 0.832 | 3.002 | 0.040 |
| $R^2 = 0.693, F = 9.011, (p = 0.40)$ |  |  |  |  |  |

**Table 2.** Relationship between track deviation and current force in ebb current.

|  | B | SE | β | t | p |
|---|---|---|---|---|---|
| (Constant) | 843.410 | 144.743 |  | 5.827 | 0.001 |
| Current Force | -434.499 | 80.076 | -0.911 | -5.426 | 0.002 |
| $R^2 = 0.831, F = 29.442, (p = 0.02)$ |  |  |  |  |  |

## 5. Conclusions

To enable a ship's captain, who has intrinsic authority to navigate the ship or the VTS operator (VTSOO) who observes multiple ships in a certain area, to make timely and correct decisions, it is necessary to predict the situation on the basis of the information provided. Since the introduction of modern VTS systems such as radio detection and ranging (RADAR), automatic radar plotting aid (ARPA), and AIS, they have continued to support navigators in decision-making. However, the task of the VTSO, such as collecting, processing, and providing more advanced information, increases the workload as the VTSO handles a large amount of ships and data. The existing position determination and pattern recognition methods use navigational data such as speed and course according to the position to decide whether there is a risk in the future position. Furthermore, route learning methods and position prediction methods using machine learning theories have been developed. However, it is necessary to study the influence of the ship's track movements in a more detailed manner, such as the

influence of currents or winds depending on the navigating area. In this study, it was verified that a specific traffic pattern appears according to the intensity and direction of the tide according to the previously proposed SVR Seaway Model, and it is suggested that these factors be considered in the navigator's decision-making in determining the course and speed. In other words, it is recognized that it is necessary to utilize weather information such as tide data in predicting a ship's track. The proposed method should be applied on the basis of the navigational area, and it can be used by applying the navigational data and environmental data of various waters. To further develop the proposed method, it is necessary to collect a large amount of data so that it can be processed through advanced big data technologies and be applied to each vessel group considering the type, size, and other particulars. These future studies are expected to improve the decision-making process, such as course change and speed adjustment. In addition, on the basis of this study, we intend to construct a position estimation and optimal route setting support system considering external forces. It is expected that the study will contribute to the prevention of marine accidents by providing valuable information to support and enhance the decision-making process of navigators. In the future, it is necessary to further study the development of a decision-making support tool suitable for various characteristic regions and ships.

**Author Contributions:** Conceptualization, H.N. and J.-S.K.; methodology, J.-S.K.; software, H.N.; validation, H.N. and J.-S.K.; formal analysis, J.-S.K.; investigation, H.N.; resources, H.N.; data curation, J.-S.K.; writing—original draft preparation, H.N.; writing—review and editing, J.-S.K.; visualization, H.N.; supervision, J.-S.K.; project administration, J.-S.K.; funding acquisition, H.N. and J.-S.K. All authors have read and agreed to the published version of the manuscript.

**Funding:** This work was supported by the National Research Foundation of Korea (NRF) and funded by the Korean government (MSIT) (No. NRF-2019RG1A1098184) and the Ministry of Education (2020R1I1A1A01060533).

**Acknowledgments:** We thank the anonymous reviewers for their careful reading of the manuscript. Their suggestions and comments helped improve and clarify our paper.

**Conflicts of Interest:** The authors declare no conflict of interest.

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
