# Peer review of "Vessel Trajectory Analysis in Designated Harbor Route Considering the Influence of External Forces"

_jmse, doi:10.3390/jmse8110860_

Round 1
Reviewer 1 Report
This paper studied support system for the decision-making of ship's navigator and VTS operator by current. This pager needs some revisions for publication.
1. Several sentences need to be corrected for publication. I recommend the English correction service by native speaker.
2. The conclusion is too long. Please include only the summary and important key points in the conclusion.
3. Please describe the final goal of your future research results.
4. This study is mainly applicable to the Yellow Sea. The limitations of the research content should be explained.
Author Response
First of all, we would like to thank you for your suggestions.
We really appreciate your efforts in helping us improve the quality of our research paper.
Please see the attachment.

Reviewer 2 Report
The paper addresses the interesting issue of analyzing specific patterns of ship trajectories in a harbor approaching area, influenced by strong currents. The proposed model could help navigators when planning a voyage in a near coastal area. However, there are few remarks to consider:
- In the introduction, the authors mentioned several times DRP, course, and speed of a ship. Somehow, I don’t see a connection of the DRP method to the proposed SVR model. More accurate would be if the paper discussed an Estimated Position, which already includes environmental influences such as wind and currents. Also, I don’t understand why the DRP method is essential here since the paper analyzes curved trajectories with many course changes. Please explain!
- SVR Seaway Model is presented very simply. Please explain the model (Fig.1) more structured and mathematically, especially I miss the information which vessel’s data are used in the model.
- Are there any papers dealing with this issue, maybe using different methods/ models, not older than five years?
Author Response

(The authors gave the same response as above.)

Reviewer 3 Report
- This paper presents a vessel trajectory analysis by considering the influence of external forces such as tidal current forces. The method improved the accuracy over the traditional methods and provides good reference for practical applications.
- The paper is written well and easy to read.
- It is suggested to provide a more detailed description of the reference area to help the reader understand the conditions of the area, such as the depth and width of the water, potential obstacles, etc.
- For Figure 5, the 24 hours are for any day in a year, or just for a specific period? How the pattern changes for different seasons?
- For Figure 10, it makes sense that there are fewer cases of deviation at “near slack condition”. Please explain why there were such big deviation at Near Slack.
Author Response

(The authors gave the same response as above.)

Round 2
Reviewer 1 Report
All requested parts of this paper have been revised. This paper is recommended for publication in this journal.